# Heatwave attribution based on reliable operational weather forecasts

Nicholas J. Leach [1,2] ✉, Christopher D. Roberts [3], Matthias Aengenheyster[1,3], Daniel Heathcote[1,4], Dann M. Mitchell [4], Vikki Thompson [4,5], Tim Palmer [1], Antje Weisheimer [1,3,6] & Myles R. Allen [1,7]

The 2021 Pacific Northwest heatwave was so extreme as to challenge conventional statistical and climate-model-based approaches to extreme weather attribution. However, state-of-the-art operational weather prediction systems are demonstrably able to simulate the detailed physics of the heatwave. Here, we leverage these systems to show that human influence on the climate made this event at least 8 [2–50] times more likely. At the current rate of global warming, the likelihood of such an event is doubling every 20 [10–50] years. Given the multi-decade lower-bound return-time implied by the length of the historical record, this rate of change in likelihood is highly relevant for decision makers. Further, forecast-based attribution can synthesise the conditional event-specific storyline and unconditional event-class probabilistic approaches to attribution. If developed as a routine service in forecasting centres, it could provide reliable estimates of human influence on extreme weather risk, which is critical to supporting effective adaptation planning.

Major advances have been made over the past decade in our ability to assess the impact of anthropogenic climate change on specific extreme weather events[1,2]. However, major challenges are still apparent when it comes to quantifying human influence in the most extreme weather events. Such events are particularly difficult to draw confident conclusions about due to the lack of historical analogues, and their often poor representation in the climate models normally used for event attribution. Given the large contribution of these exceptional events to overall socioeconomic impacts from weather, reliably quantifying how they may be affected by climate change is of considerable importance.

The key challenge extreme event attribution faces is that we cannot make direct observations of a world without human influence on the climate, so all approaches must involve some kind of modelling, either statistical[3] or dynamical[1]. Both face difficulties with the most extreme events, especially when considering the non-linear processes that often drive unprecedented events. The statistical models used in attribution studies can be inappropriate for use in the attribution of such events, given the lack of appropriately "similar" (from a physical perspective) historical samples, as demonstrated in ref. 4–6 for the case of the Pacific Northwest Heatwave. The numerical models used are often coarse (O(100 km) horizontal resolution), and poorly represent processes that are important for the development of these events, such as atmospheric rivers[7] or blocking[8].

At the same time as advances have been made in event attribution, considerable progress has been made in numerical weather prediction. The models used for weather forecasting are now considerably higher resolution than the vast majority of global climate models typically used in attribution – and are now able to provide useful forecasts at around 10 days, a lead time that has almost doubled over the past two decades[9]. This improvement is also seen in the prediction of extreme weather events, including heatwaves, which are often captured by ensemble forecasts well before they actually materialise[10–12]. Their demonstrable ability to simulate such extremes makes them

[1]Atmospheric, Oceanic, and Planetary Physics, Department of Physics, University of Oxford, OX1 3PU Oxford, UK. [2]Climate X Ltd., EC2N 2JA London, UK. [3]Earth System Predictability Section, Research Department, European Centre for Medium-Range Weather Forecasts, RG2 9AX Reading, UK. [4]School of Geographical Sciences, University of Bristol, BS8 1SS Bristol, UK. [5]Royal Netherlands Meteorological Institute (KNMI), 3731 GA De Bilt, The Netherlands. [6]National Centre for Atmospheric Science, Atmospheric, Oceanic, and Planetary Physics, Department of Physics, University of Oxford, OX1 3PU Oxford, UK. [7]Environmental Change Institute, School of Geography and the Environment, University of Oxford, OX1 3QY Oxford, UK. ✉e-mail: nicholas.leach@physics.ox.ac.uk

potentially valuable tools in quantifying the influence of climate change on high-impact weather events.

One of the key advantages of numerical weather prediction models over climate models is that we are able to confidently assess how well they are able to capture not only classes of events but also individual events[13]. It has been suggested that this ability to quantify the reliability of a model is a crucial step in improving the trust-worthiness of extreme weather attribution statements[14,15]. If a weather prediction model is able to successfully forecast an isolated extreme event for the right reasons, then this considerably improves our confidence in its ability to represent the physical processes that produced the event and external influence on these processes; ensuring that no key processes are ignored. Model reliability is a crucial property as it ensures that any estimates of changes to event probability are meaningful[16].

The other key argument for using forecast models for attribution is that they provide a framework that can synthesise the storyline[17,18] and probabilistic[1,19] approaches to attribution. While the probabilistic approach aims to determine the unconditional change in the likelihood of an event as a result of human influence based on defining the event as one of a class (for example, the annual maximum daily United Kingdom temperature[20]), the storyline – or hindcast – approach aims to determine the conditional change in magnitude of the specific event in question by examining influence on the individual physical drivers of the event. This can be done through the use of circulation analogues[21], "nudged" model simulations[22,23], or perturbed reanalysis[24]. While the storyline approach can overcome some of the problems of the probabilistic approach, including the risk of false negatives, or misattribution if the response of a specific event differs from that of the class used; it cannot provide probabilistic information about the changes in the likelihood of a particular event. This information is of clear interest to the general public and relevant to policymakers for adaptation planning. By using several different lead times, forecast-based attribution could link these two attribution frameworks, which have previously remained distinct within the literature. Short and highly conditioned lead forecasts correspond to a storyline framing, while the use of longer lead ensemble forecasts could provide probabilistic information.

At the end of June 2021, a large fraction of the Pacific Northwest region of the US and Canada experienced unprecedented high temperatures, including the cities of Portland, Salem, Seattle and Vancouver (Fig. 1). This heatwave (the "PNW heatwave") has been directly linked to many hundred excess deaths during and following it, making it the deadliest weather event on record for both Canada and Washington state[25]. The heatwave peak was observed between the 28th & 30th of June, though temperatures were still exceptionally high on the days immediately before and after this period[26,27]. A large number of local maximum temperature records were broken during this period, including the Canadian all-time record by a margin of 4.6 °C.

Based on current understanding, the heatwave arose from an optimal combination of proximal drivers[28–32]. The development of an omega block between the 23rd–27th coincided with the landfall of an atmospheric river (AR) on the 25th. Warm air was drawn up from the tropical West Pacific, heated diabatically through condensation in the river and then further heated adiabatically through subsidence: both the temperature and lapse rate at 500 hPa reached or approached record levels in the regions affected. This atmospheric heating was enhanced by soil moisture feedbacks[33,34] and high insolation at the land surface during the hottest hours of the day (Fig. 2). Given the unprecedented nature of the observed heatwave, any dynamical numerical model would need to capture all these processes, including the coupling between them, in order to produce an accurate representation of the event. This complex and unprecedented combination of processes may be why the conventional statistical models used in attribution

suggest that many of the temperature records set during the Pacific Northwest Heatwave should be impossible in the present climate[4,35,36].

Despite the observed temperatures lying far outside the historical record, the heatwave was well predicted by numerical weather forecast models such as from the European Centre for Medium-Range Weather Forecasts (ECMWF) at lead times of more than a week[37], illustrated in Fig. 1. The seasonal forecast from ECMWF captured one important aspect of the event: it predicted a thicker troposphere than average (measured by 500 hPa geopotential height) over the Pacific Northwest during the summer. A key change in the predictability of the exceptional temperatures occurred around June 21st, being the earliest point at which the penetration of the AR over land was well represented in a significant fraction of the medium-range forecast ensemble[29,30]. The success of these forecast models provides an opportunity to use them to examine the influence of anthropogenic climate change on the event as it actually occurred.

In this study, we propose a forecast-based approach to extreme event attribution. Leveraging the success of a state-of-the-art operational forecast model, we quantify human influence on the Pacific Northwest Heatwave. We perform counterfactual forecasts of the event by perturbing the initial and boundary conditions of the model in order to simulate how the heatwave might have emerged had it occurred in a cooler pre-industrial world, or a warmer future world. We then compare the counterfactual and operational forecasts to assess the impact of anthropogenic climate change on both the magnitude and probability of occurrence of the event. Although there has been some previous work into forecast-based attribution, using seasonal and subseasonal forecast models[38–43] and exploring the conceptual framework[44–46], here we use a significantly higher resolution state-of-the-art forecast model that is regularly considered to be the most reliable global forecasting system. We believe that this forecast-based approach opens the door to not only a reliable and practical operational attribution system but also to a robust way of generating projections of trends in weather-related hazards explicitly referenced to the forecasts used already by adaptation planners[47] and of relevance for a very broad audience including financial risk markets and mitigation policymakers.

## Results
### Forecast-based attribution
The methodology we develop in this study expands on the approach proposed in ref. 10. We start with an operational forecast model that was demonstrably able to simulate the event in question, as shown through a successful prediction. Here we use the ECMWF ensemble prediction system. We then perturb the boundary and initial conditions of the operational forecast (technical details in the Supplement). First, we perturb the $CO_2$ concentrations in the model atmosphere back to pre-industrial levels of 285 ppm, as done in ref. 10. Then we remove a hydrostatically balanced estimate of anthropogenic change between pre-industrial and the present day in surface and subsurface ocean temperatures, sea ice concentration, and sea ice thickness[48–50] from the initial state of the model. Perturbing the temperatures over the entire ocean depth means that we produce forecasts that are thermodynamically consistent with the changes in upper ocean heat content, in contrast to prescribed-SST approaches[51,52]. We do not alter the land surface, noting the high uncertainties in past trends for indicators such as soil moisture in this region[53–55], and evidence that, for this particular event, soil moisture anomalies were only a minor driver[56]. This design is motivated by the fact that our ocean perturbations will account for the vast majority of the total changes in heat storage of the earth system, given that ocean warming accounts for 90% of this total change[57,58]. Removing anthropogenic influence from the ocean state and reducing $CO_2$ levels produces a counterfactual "pre-industrial" forecast; we also apply identical perturbations in the opposite direction to produce a "future" forecast, in which the ocean

state and $CO_2$ levels of 615 ppm correspond to approximately twice the level of global warming and radiative $CO_2$ forcing experienced at the present day.

One key aspect of this experiment design is the choice of forecast initialisation date[10,59]. This choice influences the attribution result in three (interconnected) ways: through the predictability of the event in question and the processes that drive it; through the level of conditioning on the initial state; and through the adjustment timescale of the model to the perturbed state.

Predictability is important since it ensures that we are performing a genuine attribution of the specific event we are interested in − and using a model that is able to simulate it accurately. It is not only vital that the model is able to reliably predict the event[60], but also important that this predictability does not fundamentally change when the model

is perturbed. If the predictability is significantly altered by the perturbations imposed, then it is possible that any attribution results are just a consequence of the chaotic nature of the earth system, rather than being genuinely attributable to human influence on the event. Of course, it is possible that there may be a physical basis for such a change in predictability, and if care is taken in exploring such a basis then robust attribution statements may still be possible. However, in this study, we use the separate "pre-industrial" and "future" forecast experiments to ensure that we are not misattributing chaotic changes in predictability to human influence. We find that despite the large impulse applied by the perturbed initial state upon forecast initialisation, the predictability of the heatwave is remarkably stable on all lead times examined. Further, the key attributable changes found in the perturbed forecasts are consistent with the canonical response to

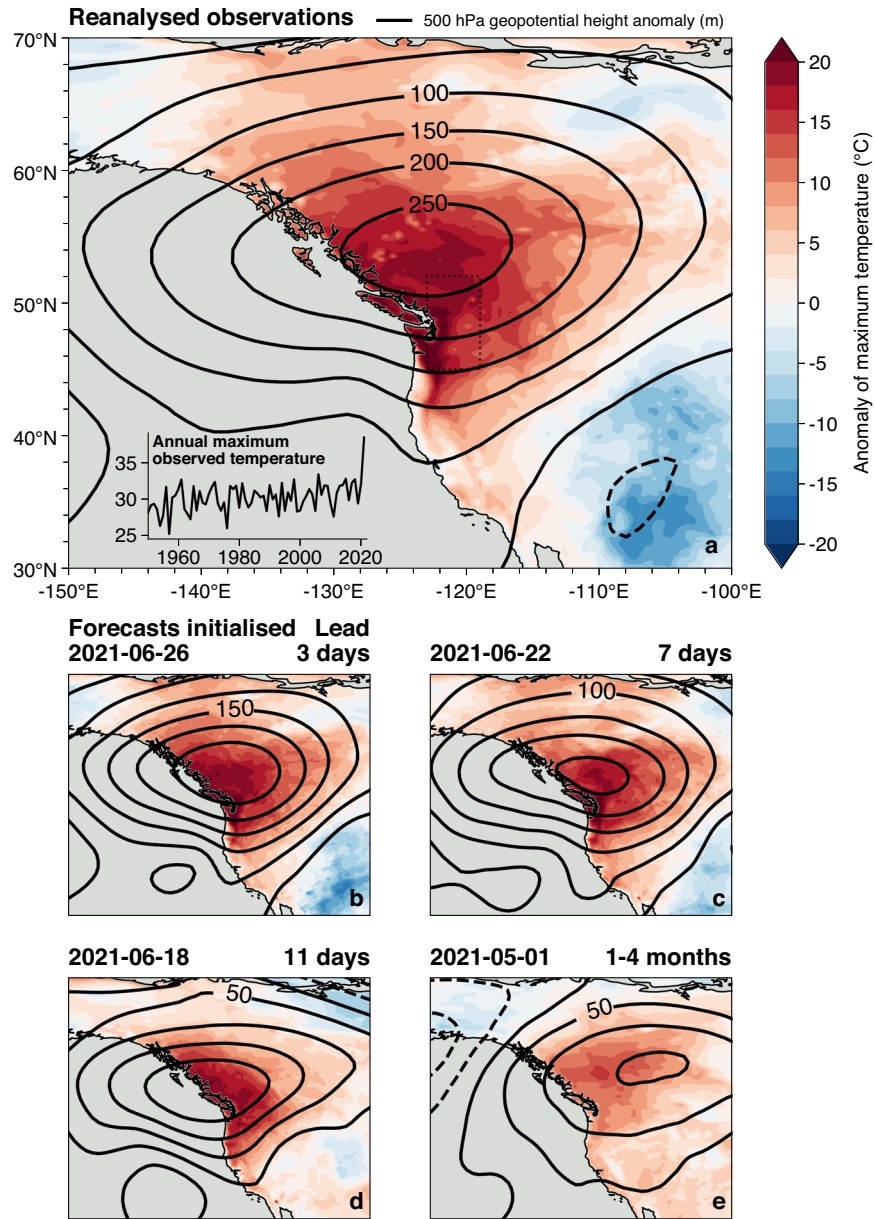

**Fig. 1 | Features and forecasts of the Pacific Northwest heatwave. a** Daily maximum surface temperature anomalies on 2021-06-29, the day of the peak heat during the heatwave within the region enclosed by 45–52 N, 119–123 W (indicated by the dotted rectangle). Solid black contours show the 500 hPa geopotential height anomaly averaged over 26–30th June 2021. Data are from ERA5 reanalysis[82]. Inset: timeseries of the annual maximum of daily maximum temperatures for the same dotted region. **b**–**e** As above, but taken from the ensemble member within the forecast initialised on the date displayed that predicted the nearest temperature to the reanalysis within the dotted region. We note that the timing of the peak of the heatwave in the various ensemble members shown does not necessarily coincide with the timing in reality.

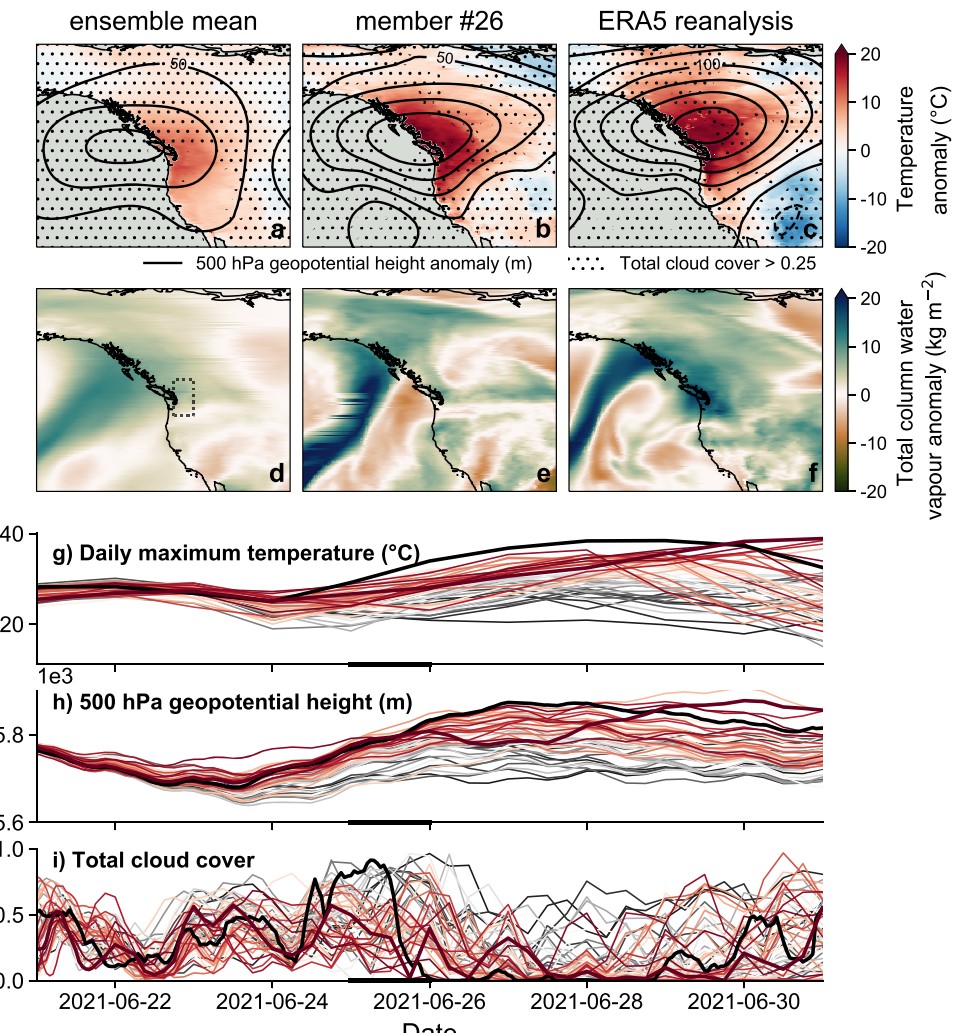

**Fig. 2 | Drivers of the Pacific Northwest (PNW) heatwave and their predictability in the operational forecast initialised 2021-06-18 (11 days). a–c** temperature anomaly fields for the PNW heatwave in the ensemble mean, nearest member and reanalysis. Solid black contours indicate 500 hPa geopotential height anomalies and stippling indicates regions with total cloud cover greater than 25%. **d–f** Mean total column water vapour anomalies on the 25th of June. The study region of 45–52 N, 119–123 W, over which fields are aggregated into time series, is indicated by the dotted rectangle. The anomalies shown are calculated relative to the 2001–2020 period. **g–i** Ensemble time series plumes of daily maximum temperatures, total column water vapour and total cloud cover respectively. The solid black line shows the reanalysis time series and the thick solid line shows the nearest member. The colour of each line indicates the rank of that ensemble member in terms of the peak temperature simulated during the heatwave period (dark grey = coolest, dark red = warmest). The solid black bar on the time axis of each panel indicates the averaging period used for the total column water vapour maps.

global warming. These include a thickening of the lower troposphere[61] and increased tropospheric water vapour[62] in the future forecast; and vice versa in the pre-industrial forecast. As such, the perturbations have not altered the forecasts in such a way that they produce "different" weather, and we can compare our forecasts to estimate the influence of anthropogenic global warming on the Pacific Northwest heatwave. This is consistent with previous work[10,59], but is not guaranteed to be the case for every weather event.

The other experimental feature we discuss in detail here is the adjustment timescale of the perturbed forecasts. Since we only perturb the ocean and sea ice initial conditions, the atmosphere and land surface are left free to adjust to the new climate state. Although the relevant atmospheric adjustment timescales are short, of order days[63], at the similarly short lead times of medium-range forecasts this adjustment may impact the attribution results. This is the reason why previous initialised attribution experiments either consider the first month to be a "spin-down" period and discard it[52]; or perturb these

additional fields using changes estimated from free-running climate model experiments[4,41]. However, such unconditioned perturbations are not necessarily consistent with the state of the atmosphere at the forecast initialisation time. The use of unconditioned perturbations therefore introduces similar adjustment issues that may be smaller in magnitude than those present in our experiments, but that are also far harder to quantify. Here we allow the model to evolve freely to the initial and boundary condition perturbations imposed. This results in smaller estimated attributable changes at shorter lead times as the atmosphere adjusts to the new climate state. However, we find that the local response of the PNW heatwave varies linearly with the global mean land temperature response across the experiments performed at various lead times ranging from three days to two months. We can therefore estimate what the total attributable impact would be if the model were allowed to fully adjust by scaling the estimated local heatwave response by the present-day observed attributable global mean land warming of 1.6 °C[64]. We use global mean land warming as

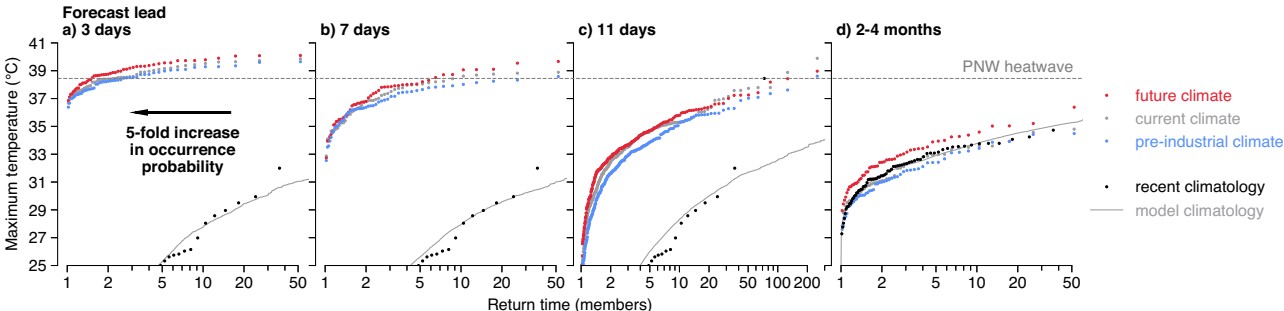

**Fig. 3 | Return-time diagram of the Pacific Northwest (PNW) heatwave in the operational and counterfactual forecast ensembles. a–d** diagrams for the forecast ensembles initialised at the lead given in the panel titles. Red, grey and blue dots indicate empirical return-time plots based on the ensemble members of the future, current and pre-industrial forecasts respectively. The dashed grey line shows the temperature threshold observed during the PNW heatwave. The black dots indicate the recent climatology, based on detrended ERA5 reanalysis over 1950–2020. The solid grey line indicates the model climatology estimated using detrended hindcasts over 2001–2020 for the medium-range forecast and using detrended and bias-corrected hindcasts over 1981–2020 for the seasonal forecast. The arrow in the left-hand panel indicates, for illustration, the displacement along the log-scaled x-axis equivalent to a 5-fold increase in occurrence probability.

this is more independent of the ocean state perturbations imposed than ocean warming. We note that this linear relationship between local extreme and global average responses, shown in Fig. S3, is consistent with previous work[65]. The shorter the lead time, the further from "full" adjustment, and the larger the scaling factor required. To further justify our experiment design, we have performed a perfect model large-ensemble experiment within a less computationally intensive coupled atmosphere-slab ocean model. This experiment demonstrates that such scaling can provide accurate and consistent estimates of the attributable signal across a range of lead times and corresponding levels of global warming. We give further details in the supplement. We note that although such a linear scaling appears to be appropriate for this particular event, it is not necessarily the case for all events – and in particular those whose attributable anthropogenic signal is due to non-linear dynamical drivers which may vary with lead time, as opposed to thermodynamic drivers[21]. In the next paragraph, we discuss the longest-lead experiments performed using seasonal forecasts, in which the atmospheric adjustment has the least impact on the estimated attributable response.

### Seasonal forecast experiments of the heatwave

The 51-member ECMWF seasonal forecasts initialised on the 1st of May were able to successfully predict the elevated large-scale temperature and geopotential height anomalies (Fig. S7). This successful prediction demonstrates that this model is fit-for-purpose, and we can therefore use it to assess human influence on the heatwave as a result of influence on the large-scale drivers[60]. On these timescales, we do not expect the forecast model to predict the precise timing of the heatwave, and so use the peak heatwave over the entire summer period in each ensemble member. We find that the attributable warming, estimated as half the difference between the pre-industrial and future forecasts to maximise the signal-to-noise ratio and scaled by the coinciding global land warming level at the time of each heatwave, is 1.2 °C [0.8, 1.6]. However, despite the successful forecast, on these seasonal timescales, the model is unable to completely capture the specific sequence of events that led to the unprecedented temperatures observed during the PNW heatwave[28–32], and so none of the members reach or exceed the temperatures observed. In order to include the proximal drivers in our assessment of human influence on the heatwave, we move to shorter lead-time experiments, which were demonstrably able to predict and accurately represent the heatwave as it actually occurred.

### Medium-range forecast experiments of the heatwave

The heatwave was well predicted by the ECMWF ensemble forecast, with the observed magnitude just captured within the ensemble from a lead time of 11 days. By a lead of 7 days, the atmospheric river location

and inland penetration, and corresponding peak temperatures were captured in many of the ensemble members[29,37]. At both of these leads, the key proximal drivers including low cloud cover and soil moisture were captured in addition to the large-scale dynamical drivers, illustrated for the 11-day lead in Fig. 2[30,34], and for the other leads in Figs. S5 and S6. We additionally consider the 3-day lead forecast, which is very highly conditioned on the synoptic drivers of the event; with a number of the key drivers effectively prescribed in the initial conditions. At this lead, the attribution experiments we perform could be considered analogous to storyline attribution framing. The remarkably successful medium-range forecasts allow us to use the forecast-based approach to attribution to examine human influence on the PNW heatwave, taking into account the specific physical processes that led to the exceptional heat – reinforcing confidence in the seasonal forecast attribution. We find that the attributable warming ranges from 0.7 [0.1, 1.3] to 1.5 [1.3, 1.7], in the 11- and 3-day forecasts respectively. In the next section, we synthesise the seasonal and medium-range results and estimate the attributable changes in the probability of the PNW heatwave.

### Synthesis

We find that the intensity of the heatwave is reduced in the pre-industrial forecasts for all lead times (Fig. 3). Synthesising the (scaled) estimated attributable signal across all lead times, we find a best-estimate anthropogenic impact on the heatwave intensity of 1.3 °C [0.7, 1.6] for a current level of anthropogenic warming of 1.25 °C[66]. While this is smaller than the total anomaly of 7 °C over the climatological annual maximum, it is important to stress that the impact of any additional warming is strongly non-linear[67], such that a 20% increase in heatwave magnitude may result in a much greater than 20% increase in heatwave impacts. While we do not model these impacts explicitly here, our detailed simulations would provide the information necessary to do so.

We quantify the attributable change in probability due to anthropogenic global warming using relative risk[68], estimating the probability of observing a temperature at least as extreme as the peak of the observed 2021 heatwave using an appropriate extreme value or tail distribution, and then shifting this distribution by the attributable change in intensity for each lead time. Our results are consistent with a linear relationship between log probabilities and the coinciding global land warming level independent of forecast lead time, which is consistent with the hypothesis that human influence on climate is primarily manifest in the development of the heatwave from precursor conditions, rather than through affecting the probability of occurrence of those precursor conditions themselves. Further research into human influence on weather precursors is clearly called for. If we

account for this adjustment by scaling log probabilities by the current global land warming level of 1.6 °C, just as we did with the estimated attributable signal, we find a best-estimate relative risk of a factor of 8 times [2, 50] considering all lead times, or analogously a fraction of attributable risk (FAR) of 0.9 [0.5, 0.98]. We find that the relative risk tends to decrease towards a constant value as the lead time increases, resulting from the reduced level of conditioning[20]. Our estimate of the relative risk using the most unconditioned lead time, the seasonal forecast, is a factor of 5 [2, 9]; or a FAR of 0.8 [0.6, 0.9]. This relatively unconditioned seasonal timescale result would be the most comparable to previous probabilistic attribution estimates[35] − though using a model that is demonstrably able to simulate the event in question[60].

Using the current rate of global warming over land[66] we can further estimate that the probability of observing an event at least as warm as the 2021 Pacific Northwest heatwave is doubling every 20 [10, 50] years, and will continue to do so unless the rate of global warming decreases. Given the length of the historical record and our estimated change in probability over this period, such an event would be associated with a multi-decade to multi-century return period at the present day, thus making this doubling time extremely relevant for adaptation planning.

## Discussion

The results presented here provide strong evidence of the impact of climate change on a specific extreme event, based on a model that has been demonstrated unequivocally to be able to simulate the event in question through a successful medium-range forecast. Our estimates of relative risk are lower than some previous climate-model-based estimates[35], albeit are not entirely incompatible within the context of the associated uncertainties and the fact that the raw estimates from our experiments would represent a lower bound on the impact of climate change on the heatwave due to the adjustment to perturbed initial conditions (as was the case in ref. [10]), though we have accounted for this by relating attributable local- to global-scale signals. One reason for this discrepancy is that our model (unlike a typical climate model) is capable of simulating the multiple physical factors that contributed to the heatwave that occurred, so we are not relying on the extrapolation of distributions from physically dissimilar events. Another reason is our imposed initial condition perturbations do not include the total sum of human influence on the climate, and so the impact of some non-linear interactions between different climate system components may not be fully realised. It is known that land surface feedbacks are important in the development of extreme heatwaves[69], although possibly only modestly so for this particular case[56], and is therefore plausible that if we had removed the influence of anthropogenic climate change from the initial land state in addition to the ocean state, the resulting attribution statement might have been stronger. The similarity of the relative risk estimates from the seasonal and longest medium-range forecasts, together with the finding that heatwave magnitude scales with global land temperature anomaly, both indicate that the impact of human influence on the probability of occurrence of precursor conditions at the time of forecast initialisation is relatively small, but there may be important non-linear interactions that would also result in a stronger attribution.

Nevertheless, we argue that the forecast-based methodology presented here represents an important advance not only towards operational attribution but also attribution in general. Rather than relying on multiple lines of evidence that would each be unsatisfactory in isolation, here we have presented a single adequate line. The key to the adequacy of the result is the ability of the model used to represent the event in question, demonstrated through successful prediction. This not only means that we have increased confidence in the model's response to external forcing[60,70], but also that the analysis is a genuine attribution of the specific event that occurred (rather than a mixture of events that share some characteristic like extreme temperatures, but

differ in other important meteorological aspects). Forecast-based attribution provides many of the advantages of the storyline approach to attribution, but can still be used to provide quantitative estimates of the changing probability of extreme events with climate change, as demonstrated here. Furthermore, it can synthesise the storyline and probabilistic approaches to attribution within a single framework. The use of an operational weather forecast model demonstrates how this approach could be easily adapted to provide an operational system for attribution in real-time (or potentially even in advance[41,59]). Such a system would involve re-running operational forecasts with perturbed initial and boundary conditions as in the counterfactual forecasts we have presented here[71].

We suggest that as well as these broad arguments for forecast-based attribution in general, our study and experiment design provide several developments over prior work[38–43]. Here we have used the ECMWF ensemble prediction system, widely regarded as the most reliable global ensemble forecast model available. It is used by weather centres to provide forecasts or drive regional models across the globe. This presents a real opportunity for an operational attribution service: to use a well-tested global model that is already used in a genuinely global rather than regional context. The operational system we have used in this work is run at a resolution of 18km, significantly higher than previous forecast-based attribution work. This increased resolution provides two benefits: it allows attribution to be carried out on spatial scales that are more locally relevant and it improves the representation of potentially key physical processes. This has meant that we have been able to focus on the short-term peak of the heatwave in question, rather than coarser temporal and spatial event definitions as have previously been used. In addition, the approach we have used for generating perturbations does not rely on expensive and potentially highly biased climate model simulations, whilst ensuring that the new initial ocean states are hydrostatically balanced.

However, there remain a number of ways in which the forecast-based approach explored here should be further developed. A clear complementary experiment would be to include additional perturbations to the initial conditions, in order to reduce the model adjustment to the new climate. This would require estimating attributable changes in the land surface and atmosphere, such as in refs. [41], [72]. While such changes to the land surface will likely not significantly affect the predictability in such a way as to confound any attribution results, the atmosphere will require more care in order to ensure that the applied perturbations are consistent with the specific state at the time that the model is initialised, for example by introducing perturbations during the data-assimilation procedure, or by iteratively estimating this delta through successive forecasts. We also could include additional secondary factors in the perturbed boundary conditions, such as the effect of anthropogenic aerosols and other greenhouse gases, in addition to the changes to the $CO_2$ concentration made here. Another key development will be to apply this approach within alternative forecast models to check the sensitivity and consistency of attribution results to the model used − our use of a single model is a limitation here. With all of these developments, this methodology could provide a robust and generic approach to estimate the changes in extreme weather risk operationally[71]. Finally, we suggest that the uncertainty in the perturbations applied to the initial state could be more fully explored, as has been done in previous attribution experiments[73].

This study focused on the attribution question, but the same forecast-based methodology could be applied to produce projections designed to inform climate change adaptation[74]. Analogous to our "future" counterfactual forecast, which we used here to check the linearity of the climate change response and increase the attributable signal-to-noise ratio, perturbations consistent with specific levels of global warming could be applied in order to, for example, simulate individual extreme events as if they occurred in a world of 2 °C or other policy-relevant targets. Such simulations of potential future extremes

could be used to test the limits of regional adaptation in a targeted manner based on impactful events that have already occurred[47], complementing other approaches such as ref. 75, which was designed to produce a rich set of different extreme events rather than specific "grey-swan" type events. The use of forecast models is a key advantage as these are already used in a wide variety of contexts such as assessing weather-related risk for reinsurance[76], and as part of impact-modelling chains for issuing real-time hazard warnings[77].

In this study, we have used a numerical weather forecast-based approach to determine the contribution of human influence to a specific unprecedented extreme event. We used a reliable state-of-the-art coupled operational weather forecast model that was unequivocally able to simulate the event in question, demonstrated by a successful prediction. Our forecast-based approach maintains consistency with the measured changes in upper ocean heat content, unlike many previous approaches, and can synthesise the storyline and probabilistic approaches to event attribution in a single framework, keeping the event specificity of the storyline approach while still providing meaningful estimates of the changing risk of the extreme in question. One key avenue for further research lies in applying state-dependent perturbations to the initial atmospheric conditions; since in this work, the model adjusts to the perturbed ocean conditions continuously throughout the forecast. Although we have accounted for this adjustment in our analysis here, the interpretation of the results would be simplified if it could be done during the experiment itself. Given that it is increasingly clear that we need to go beyond the meteorology of event attribution, and into the societal impacts[67,78], we suggest that our approach would be particularly well placed to advance this agenda, especially in the context of extremes in a future climate.

## Methods
### Event definition
How the extreme event of interest is quantified - the event definition - is a key methodological decision that must be made in extreme event attribution studies. A significant amount of previous work has shown the impact of the event definition on the quantitative outcome of the analysis[79–81]. In this study, we use a definition consistent with a previous attribution study of the PNW heatwave[35] to allow for a comparison between our forecast-based approach and their probabilistic statistical and climate-model-based approach.

We first average maximum temperatures over the region enclosed by 45–52 N, 119–123 W (indicated by the dotted rectangles in Figs. 1 & 2). For the event as observed in the ERA5 reanalysis[82] we then take the peak temperature recorded during the heatwave, which occurred at 00 UTC on 2021-06-29. For the event as simulated in the medium-range forecast ensemble members, we take the peak temperature that occurred between the 26–30th of June, the period over which the heatwave occurred in reality. For the event as simulated in the seasonal forecast ensemble members, which we would not expect to predict the precise timing of the heatwave, we take the peak temperature over the full summer season. The differences between the event definitions of the medium-range and seasonal cases lead to the discrepancies in the climatologies shown in Fig. 3.

### Experiment details
**Model details.** The medium-range experiments we have performed use the version of the IFS EPS that was operational at the time of the PNW heatwave, CY47R2[83]. The forecast model atmosphere is run at a resolution of Tco639 (18 km) and has 137 vertical levels. The atmosphere is coupled to a 0.25-degree wave model[84], 0.25-degree sea ice model[85], LIM2, and 0.25-degree ocean model[86], NEMO v3.4, with 75 vertical levels (ORCA025Z75 configuration). We maintain the same number of ensemble members as the operational system,

51, throughout our experiments, with the exception of the 11-day lead initialisation, where we expand the ensemble size to 251 members to increase the statistical confidence in our results from this ensemble.

The seasonal experiments are performed with ECMWF`s operational seasonal forecasting system, SEAS5[87]. This uses IFS CY43R1[88] at a horizontal resolution of Tco319 (36 km) with 91 vertical levels. The seasonal configuration of IFS CY43R1 is coupled to a 0.5-degree wave model, LIM2, and NEMO v3.4 in the ORCA025Z75 configuration. We maintain the same number of ensemble members as the operational system, 51, throughout our experiments.

**Simulation setup.** Our experiments all use the exact operational setup (model configuration and initial conditions) as their base. To this setup, we:

1. Change the CO2 concentrations used to a "pre-industrial" level of 285 ppm, and a "future" level of 615 ppm. These represent the same fractional change in opposite directions from the present-day concentration of 420 ppm used in the operational forecast system.

2. Subtract (for the pre-industrial forecast) or add (for the future forecast) a perturbation of the estimated anthropogenic influence on the ocean state since the pre-industrial period from the initial conditions of the forecasts (through the ocean restart files). The estimation of this perturbation is described below. We use estimated perturbations for 3D temperature, sea ice concentration, and sea ice thickness.

3. Check the sea ice fields for unphysical values. In the perturbed restarts, we ensure that sea ice concentration does not exceed 1 or subceed 0. We ensure that sea ice thickness does not subceed 0. Values outside these bounds are set to their nearest bound. Finally, we set sea ice thickness to 0 where sea ice concentration is 0, and vice versa.

4. Modify ocean salinity such that in-situ ocean density is preserved following the 3D temperature perturbation as calculated using the equation of state from the forecast ocean model. The salinity compensation is achieved to machine precision using a simple gradient descent algorithm. The resulting coupled forecasts are thermodynamically consistent with the imposed ocean heat content anomalies without any adjustments to the initial ocean circulation, mixed layer depths, or horizontal pressure gradients. Importantly, and unlike uncoupled forecasts constrained by specified sea-surface temperatures, there are no infinite sources or sinks of heat in the resulting counterfactual forecasts. This approach is justifiable in shorter-range forecasts as there is no direct influence of salinity on the overlying atmosphere. This assumption may eventually break down at lead times comparable to ocean advective processes, for which there could be indirect feedback on the atmosphere associated with salinity-driven changes in the ocean state. Nevertheless, this approach works well for the medium-range and seasonal forecasts described in this study.

The perturbations used are computed using an optimal fingerprint analysis[66,89,90]. We first calculate the Anthropogenic Warming Index (AWI) using anthropogenic and natural radiative forcings from AR6[91] and the HadCRUT5 global mean surface temperature dataset[92]. The AWI provides us with a plausible estimate of the fingerprint of anthropogenic influence on other climate variables[89]. For each perturbed variable, we then regress the observed time series at each gridpoint onto the AWI, using the following data sources:

- Sea ice thickness: ORAS5 (1958:2019)[50]
- Sea ice concentration: ORAS5 (1958:2019)[50]
- Sea-surface temperature: HadISSTv1.1 (1870–2019)[49]
- Subsurface temperature: WOA18 (1950–2017)[48]

We then scale the computed regression coefficients at each point by the change in AWI between the pre-industrial period of 1850–1900 and 2019 to produce our final estimated perturbations. The sea surface and zonally and globally averaged temperature profiles are shown in Fig. S2.

Finally, we combine the sea-surface and subsurface temperature perturbations. We did not use a subsurface temperature dataset in isolation since observations of the sea-surface temperature are considerably more abundant in the early 20th century than observations of subsurface temperatures, and since the temperatures at and near the surface are likely to be the most important for the medium-range forecasts performed, we leveraged the additional information contained in observed sea-surface temperatures. We combine the two by relaxing the sea-surface perturbation towards the subsurface perturbation using a relaxation depth scale of 60 m (the surface auto-correlation scale in WOA18).

We note that estimation of the perturbation, and in particular the subsurface temperatures, is associated with considerable uncertainty due to the lack of observations in the pre-ARGO era. Here we have used a single best-estimate perturbation due to constraints on the available computational resource, but to account for this uncertainty an ensemble of perturbations could be applied[73]. A possible way in which such an ensemble could be derived would be to apply optimal fingerprinting to an ensemble of coupled climate models. We note that we avoided using individual or ensemble-averaged climate models to estimate the ocean state perturbations, as done in other studies, due to the significant biases present in the current class of these models, especially in the subsurface.

**Bias correction of seasonal forecast ensembles.** Climate drift can be an issue in the use of coupled seasonal forecast models[93]. We find a non-negligible drift in the daily maximum temperature SEAS5 forecast ensemble initialised in May over the PNW region. This drift results in a positive temperature bias that grows with lead time. Hence using the raw model output in our analysis would overestimate the probability of the PNW heatwave.

To account for the drift, we perform a simple bias-correction procedure on the seasonal forecast ensembles, informed by comparing the SEAS5 hindcasts over 1981–2020 with ERA5 reanalysis data over 1950–2020 (using the full-time period that data is available and excluding the year of the event, 2021). We do this in three steps:

1. Remove the attributable forced trend from both the reanalysis and hindcasts by regressing mean JJA daily maximum temperatures onto the AWI[89].
2. Remove the drift from these detrended hindcasts, estimated by averaging the hindcasts for each lead time over all years and ensemble members, subtracting this from the corresponding reanalysis average over all years, and then regressing this time series onto the (linear) lead times, producing a linearly lead-time dependent drift correction[93].
3. The drift-corrected hindcasts still exhibit a positive bias during periods of extremely high temperatures. Hence we finally remove the remaining mean bias in annual maximum temperatures in the hindcasts compared to reanalysis.

We apply this bias-correction procedure to both the seasonal hindcasts shown in Fig. 3 and used to estimate the return time of the event and to the operational and perturbed seasonal forecasts of the 2021 summer. Figure S4 shows the results of this bias-correction procedure, following Thompson et al.[94].

We note that validation of the bias-correction procedure on the SEAS5 distribution of annual maximum temperatures (TXx) is challenging due to the unprecedented nature of the 2021 event. If we perform an analysis of the higher-order moments of the SEAS5 and

"observed" (ERA5 reanalysis over 1950–2020) distributions of TXx[94], we find that the bias-corrected ensemble tends to have larger values of higher-order moments than the observed time series. However, if the 2021 event is included in the observed distribution, then the opposite is found, due to the large impact of such an outlier on these moments. This sensitivity to inclusion/exclusion of the 2021 event, demonstrated in Fig. S2, is why we have opted to perform a simple but physically motivated bias correction rather than a more complex statistical correction such as a quantile map.

### Statistical methodology

**Intensity changes.** We calculate changes in intensity by first selecting the members from the operational ensemble within the nearest quintile to the event. We then calculate the difference between the average of these members in each ensemble. For the three longer leads, this is effectively the difference between the averages of the uppermost quintile of the two ensembles.

**Risk changes.** We calculate the risk ratio by fitting a straight line on a return-time diagram (i.e. an exponential tail) to the nearest quintile of either the operational ensemble (for the other two medium-range leads) or the model climatology (for the seasonal lead since the tail of the operational ensemble lies considerably further below the event threshold than the tail of the much larger model climatology). We do this, as opposed to more traditional extreme value (EV) analysis because several of the ensembles are not well represented by conventional EV distributions, and have generally heavier tails than estimated by e.g. likelihood-maximising Generalised EV distributions. Many of the required assumptions of EV theory would be violated here also. We note that where the event threshold lies in the extreme tail of the ensemble, the tail properties of any fitted parametric distribution project considerably onto the estimated probability of the event. Hence to avoid any undue assumptions on the tail shape, we fit a straight line on a return-time diagram such as Fig. 3 (assuming an exponential, Gumbel, tail) to the nearest quintile.

After fitting an appropriate distribution, we then shift the location of this distribution by the estimated attributable warming. We then calculate the probability of observing an event at least as intense as the PNW heatwave (the dashed line in Fig. 3) in the original distribution and the shifted distribution. The risk ratio is the ratio of these two probabilities ($P_{current}/P_{shifted}$).

Throughout, confidence intervals are calculated using a 10,000-member non-parametric bootstrap with replacement. Numbers in square brackets represent a likely confidence range of 17–83%.

## Data availability

All experimental data generated in this study are available with anonymous access from ECMWF MARS under experiment IDs b2m8, b2mc, b2mg, b2mh, b2mk, b2ml, b2mp, b2mq and b2mr within the UK experiment class. Processed data required to reproduce the figures are available at https://doi.org/10.5281/zenodo.10728125.

## Code availability

Code used to produce the figures is available at https://doi.org/10.5281/zenodo.10728125.

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

## Acknowledgements

We thank Paul Dando, Robin Hogan and Andrew Dawson for their help with setting up IFS for our experiments. The results contain modified Copernicus Climate Change Service information [2022]. Neither the European Commission nor ECMWF is responsible for any use that may be made of the Copernicus information or data it contains. This work was supported by the Natural Environment Research Council (NE/L002612/1) (N.J.L), the European Union's Horizon 2020 project FORCeS (821205) (N.J.L. and M.R.A.), the Natural Environment Research Council project EMERGENCE (NE/S005242/1) (D.M.M. and V.T.), the European Research Council Advanced Grant ITHACA (741112) (T.P.), the European Union's Horizon 2020 project European Climate Predictions (776613) (A.W.), and the European Union's Horizon Europe research and innovation programme (101081460) (A.W.). Acknowledgement is made for the use of ECMWF's computing and archive facilities in this research under the special project *spgbleac*.

## Author contributions

Conceptualisation: N.J.L., T.P., A.W., M.R.A.; methodology: N.J.L., C.D.R., A.W., M.R.A.; software: N.J.L., C.D.R., M.A.; formal analysis: N.J.L., M.A., D.H.; investigation: N.J.L.; data curation: N.J.L.; writing - original draft: N.J.L.; writing - review and editing: N.J.L., C.D.R., M.A., D.M.M., V.T., T.P., A.W., M.R.A.; visualisation: N.J.L., M.A.; supervision: A.W., M.R.A.

## Competing interests

N.J.L. works part-time for Climate X Ltd., a company that provides climate risk analytics to the financial sector. The remaining authors declare no competing interests.
