## [Peer Review File · Nature Communications]

Heatwave attribution based on reliable operational weather forecastsEditorial Note: This manuscript has been previously reviewed at another journal that is not operating a transparent peer review scheme. This document only contains reviewer comments and rebuttal letters for versions considered at *Nature Communications*.

REVIEWER COMMENTS

Reviewer #1 (Remarks to the Author):

The authors provided substantial revisions to address most of my previous concerns. Given the short-lived nature of the Pacific Northwest heatwave and the potential importance of atmospheric memory, I still have the concern on whether the proposed model experiments could provide an accurate estimation of the global warming effect on this specific heatwave event. I hope this concern can be somehow further addressed although it might be difficult. As far as I understand, the proposed experiments by perturbing ocean conditions and CO₂ forcing do not provide essential differences from existing non-operational studies.

Reviewer #2 (Remarks to the Author):

A review of "Heatwave attribution based on reliable operational weather forecasts" by Leach et al. submitted in Springer Nature Communications

The authors study the 2021 Pacific Northwest (PNW) heatwave event and its drivers in the operational ECMWF ensemble medium-range and seasonal prediction systems and assess the impact of climate change on this event using a set of perturbation experiments in terms of its return time in the present and counterfactual ensemble forecasts.

The research topic is interesting and the manuscript is generally well-written. This study uses the forecast-based attribution method, which is relatively less explored compared to the probabilistic approach. The analysis and interpretation are found to be clear, and the results support the arguments with some useful information that may contribute to climate attribution and operational community. However, the authors claim to do this for the "first time" in the abstract (L8) and introduction (L95-96) sections, which is misleading. A set of studies has already applied forecast-based extreme event attribution in recent times. Additionally, the perturbation techniques, as the authors also acknowledged, could be improved to reduce the impact of climate adjustments due to the initial and boundary conditions alteration before fully implementing this method operationally. A few comments are included below that authors should consider in improving the manuscript.

Major points:

1. The initial perturbations are added only to the ocean state and the atmosphere-land systems are allowed to freely adjust to these changes in the forecasts. Many of these adjustments through the air-sea interaction take longer than the predictability of the drivers associated with the PNW heat wave. In that case, the attribution results at 11-day lead times may still be weaker than expected. Although the authors discuss the atmospheric adjustment timescales, this could be a major weakness of this strategy

for its operational use. Your comments on this.

2. The predicted surface temperature anomalies are weaker than observed at the 11-day lead time, even in the best-performing member #26 (Fig.2). So, the attribution estimates based on this forecast are expected to be weaker. Although the 3-day lead forecast is close to reality, this can't be used for the O-A-land coupled system adjustment. A discussion on the predictability vs the adjustment time should be added to the conclusion section. If the authors can find a remedy for this demerit of the forecast-based attribution method, it would add value to this study.

3. Some of the cited studies (e.g., 27-31) applied forecast-based attribution to extreme events using the operational O-A coupled prediction system. Recently, Abhik et al. (2023) carried out a heatwave event attribution in Australia using an operational O-A coupled S2S ensemble prediction system and estimated the role of GHG increase in that extreme heatwave event by perturbing the initial and boundary conditions, which seems to be quite relevant to this study.

Abhik, S., Eun-Pa Lim, Pandora Hope, and David A. Jones. Multiweek Prediction and Attribution of the Black Saturday Heatwave Event in Southeast Australia. *Journal of Climate* 36, 19 (2023): 6763-6775.

4. L95-96: Other than using a high-resolution model (and different perturbation techniques), what are the key differences between this study and the studies mentioned in comment#3? A brief discussion should be added on the uniqueness of the present study.

5. If the perturbation is also added to the atmospheric initial conditions, does it increase the risk estimates provided in Fig.3 and the FAR score?

Minor points:

1. L73: "UK" should be fully mentioned at its first use. The same applies to "ECMWF".
2. L109-110: What was the range of the max temperature anomalies?
3. Fig.1: Is the top panel showing surface temperature anomalies during 26-30 Jun 2021 as the geopotential height anomaly? Please specify.
4. Fig.1 (Inset): Is it the annual mean maximum temperature or the annual maximum value of the daily maximum temperature?
5. How many ensemble members out of the grand ensemble are capable of reaching close to reality?
6. If the plotted surface temperature and geopotential anomalies indicate 26-30 Jun 2021 conditions as in the top panel of Fig.1, the lead times for the 1 May forecast should be less than 2 months.
7. P15: remove space between numbers and the deg-C sign.

Response to reviewers

We have written point-by-point responses to each review comment. The original reviews are in italics - and our responses are preceded by “%%”.

Reviewer #1 (Remarks to the Author):

The authors provided substantial revisions to address most of my previous concerns. Given the short-lived nature of the Pacific Northwest heatwave and the potential importance of atmospheric memory, I still have the concern on whether the proposed model experiments could provide an accurate estimation of the global warming effect on this specific heatwave event. I hope this concern can be somehow further addressed although it might be difficult. As far as I understand, the proposed experiments by perturbing ocean conditions and CO2 forcing do not provide essential differences from existing non-operational studies.

%% Thank you for these comments and your previous review. We certainly share your concern, and for this reason have highlighted this limitation and corrected for it. We also note that other existing experiment designs (such as applying multi-model mean estimates of the anthropogenic influence on the atmosphere) also fail to address this potential issue, but in a way that is more difficult to identify. This limitation of the current experiment design is something that we are actively working to address, though it certainly does represent a practical challenge - and for this reason we believe it lies outside of the scope of this study. On your final point about the differences from existing work, while the underlying design itself (ie. perturbed weather forecasts) is by no means a new idea, we argue that several crucial aspects of our study set it apart from previous work: 1) The emphasis on the use of a model that is demonstrably able to forecast the event in question; 2) the use of a state-of-the-art and extensively and constantly validated operational medium-range weather forecast model run at a higher resolution that far exceeds the models used in other studies; 3) the use of an operational model, which could in theory be leveraged as part of an operational attribution system; 4) the perturbation estimation and application methodology, which removes the need for long-running, expensive, and potentially severely biased climate model simulations. We have added text commenting on these aspects to the discussion section (340-349, 379-383).

Reviewer #2 (Remarks to the Author):

A review of “Heatwave attribution based on reliable operational weather forecasts” by Leach et al. submitted in Springer Nature Communications

The authors study the 2021 Pacific Northwest (PNW) heatwave event and its drivers in the operational ECMWF ensemble medium-range and seasonal prediction systems and assess the impact of climate change on this event using a set of perturbation experiments in terms of its return time in the present and counterfactual ensemble forecasts.

The research topic is interesting and the manuscript is generally well-written. This study uses the forecast-based attribution method, which is relatively less explored compared to the probabilistic approach. The analysis and interpretation are found to be clear, and the results support the arguments with some useful information that may contribute to climate attribution and operational community. However, the authors claim to do this for the “first time” in the abstract (L8) and introduction (L95-96) sections, which is misleading. A set of studies has already applied forecast-based extreme event attribution in recent times. Additionally, the perturbation techniques, as the authors also acknowledged, could be improved to reduce the impact of climate adjustments due to the initial and boundary conditions alteration before fully implementing this method operationally. A few comments are included below that authors should consider in improving the manuscript.

%% Thank you for your constructive comments and suggestions, they have been extremely helpful in improving the manuscript and we hope you appreciate the revised version.

Major points:

1. The initial perturbations are added only to the ocean state and the atmosphere-land systems are allowed to freely adjust to these changes in the forecasts. Many of these adjustments through the air-sea interaction take longer than the predictability of the drivers associated with the PNW heat wave. In that case, the attribution results at 11-day lead times may still be weaker than expected. Although the authors discuss the atmospheric adjustment timescales, this could be a major weakness of this strategy for its operational use. Your comments on this.

%% We fully agree with your point and believe that it is plausible that the attribution statement may have been stronger if - in particular - perturbations had been added to the land surface state in addition to the ocean. We discuss this in the first paragraph of the discussion section, noting that other work suggests that antecedent soil moisture conditions only contributed modestly to the heatwave; and that attribution of soil moisture to human influence is highly

uncertain for this region. We also note that in the seasonal forecasts (where the upper layers of the land surface will have had time to at least partially adjust), the attribution statements are still consistent with, if slightly weaker than, the shorter leads. We make this exact point over L295-302. We are actively working to address this adjustment issue, though believe that it lies out of the scope of this study due to the technical and scientific challenges involved. We also suggest within the text that it is one of the most important areas for further work (L340-349, 379-383).

2. The predicted surface temperature anomalies are weaker than observed at the 11-day lead time, even in the best-performing member #26 (Fig.2). So, the attribution estimates based on this forecast are expected to be weaker. Although the 3-day lead forecast is close to reality, this can't be used for the O-A-land coupled system adjustment. A discussion on the predictability vs the adjustment time should be added to the conclusion section. If the authors can find a remedy for this demerit of the forecast-based attribution method, it would add value to this study.

%% Your point is a very good one - it would indeed be extremely unusual for a forecasting system to forecast such a large anomaly at such a long lead time. We note that it is impressive that any ensemble members are within 1 degree of the observed event, given how far outside the historical record it lies. However, we don't entirely agree with your conclusion that this guarantees weaker than expected attribution estimates. The estimates would only be weaker if the additional physical processes that aren't present or fully realised in the longer lead forecasts are further enhanced by human influence the stronger they are. While this may be the case for precipitation extremes based on the current state of climate science (Myhre et al., 2019), the evidence for this is weaker for heat extremes, as other studies have shown that average and extreme surface temperature changes are often strongly related; and similar in magnitude (McKinnon & Simpson, 2022). It is possible that human influence on some processes would also act in the opposite direction or weaken with increasing magnitude (eg. soil moisture dryness saturating). We have shown that the model used is demonstrably able to simulate the physical processes driving the event (which we have shown is true at longer leads, even if not every ensemble member captures these processes). We would remark that this bar we have set in terms of model suitability, ie. that the model can demonstrably simulate the relevant processes, is far higher than in previous attribution studies, and is a key motivating factor for the method. We also note that - as with conventional attribution studies - the statistical methods used are most heavily influenced by samples in the tail that are the most relevant to the processes driving the event as observed. Having said this, we do agree with your suggestion to elaborate on the predictability and adjustment timescales within the conclusion, and have added in text in line (L379-383).

3. Some of the cited studies (e.g., 27-31) applied forecast-based attribution to extreme events using the operational O-A coupled prediction system. Recently, Abhik et al. (2023) carried out a heatwave event attribution in Australia using an operational O-A coupled S2S ensemble prediction system and estimated the role of GHG increase in that extreme heatwave event by perturbing the initial and boundary conditions, which seems to be quite relevant to this study.

Abhik, S., Eun-Pa Lim, Pandora Hope, and David A. Jones. Multiweek Prediction and Attribution of the Black Saturday Heatwave Event in Southeast Australia. Journal of Climate 36, 19 (2023): 6763-6775.

%% Thank you very much for bringing this relevant work to our attention - we now include a reference to it in the main text (citation #33). We have also removed the text from the abstract about novelty.

4. L95-96: Other than using a high-resolution model (and different perturbation techniques), what are the key differences between this study and the studies mentioned in comment#3? A brief discussion should be added on the uniqueness of the present study.

%% Regarding the perturbation technique, our approach removes the need for long-term climate simulations to estimate the ocean state delta, and ensures that the new initial condition is fully balanced. This is advantageous since:

- the current class (ie. CMIP6) of climate models still have significant biases in ocean surface and subsurface warming trends when compared to observations, which could heavily impact any attributable changes;
- long-running climate model simulations are computationally expensive; and,
- removing temperature and salinity deltas that have been estimated separately (as opposed to our approach of estimating temperature deltas and adjusting the salinity as required) does not ensure that the ocean density profile is maintained and could cause spurious mixing to occur once the forecast is initialised due to the non-linearity of the ocean equation of state (though we expect that any such mixing would have little or no impact on estimated attributable changes).

%% Regarding your comment on high-resolution, we believe that the key point here is that the model we used, IFS, was able to capture the study event in both absolute and relative (ie. anomaly) terms. This has not been the case in the studies mentioned, potentially due to the reduced resolution. These prior studies have also focussed on coarser scale (spatial and temporal) event definitions, possibly as a result of model deficiencies. We very much agree with this approach of not pushing a model beyond what it is capable of simulating. However, if as a community we are interested in assessing both long and short-term extremes, and in assessing impacts of these events - and we believe that we should be - then this advance towards using models that are genuinely capable of simulating the events in question is key. For these reasons, we would argue that although these experimental differences may seem trivial on paper, they are important.

%% We have added text commenting on these points to the discussion section (L324-339)

5. If the perturbation is also added to the atmospheric initial conditions, does it increase the risk estimates provided in Fig.3 and the FAR score?

%% We certainly agree with you that this is a key further experiment to be carried out and are actively working to do so! However, due to the technical challenges associated, we believe that performing this experiment merits a dedicated study and thus lies out of the scope of this work. These technical challenges include: estimation of a state-specific perturbation to be added; introduction of this perturbation into the model (as the initial atmospheric state is involved in the 4D-Var DA procedure, how best to do this theoretically and in practice in code is a challenge); and ensuring balance in the new atmosphere to avoid spurious large-amplitude gravity wave production in mountainous regions (Zarzycki & Jablonowski, 2015). However, given that we have already accounted for the adjustment of each lead time in our presented risk estimates - albeit in a relatively simple way - we wouldn't necessarily expect this additional experiment to increase them. (L340-349, 379-383)

Minor points:

1. L73: *"UK" should be fully mentioned at its first use. The same applies to "ECMWF".*

%% We have amended the text in line with your comment.

2. L109-110: *What was the range of the max temperature anomalies?*

%% Assuming that you are asking about the spatial range of maximum temperature anomalies, this is something that can be inferred from Figure 1: over the region struck by the heatwave the maximum anomalies (relative to the 1991-2020 climatology of that day of the year) were just under 24C.

3. Fig.1: *Is the top panel showing surface temperature anomalies during 26-30 Jun 2021 as the geopotential height anomaly? Please specify.*

%% This panel is showing the surface temperature anomaly at the time when the temperature enclosed by the study region reached its maximum value; whereas the geopotential height anomaly is indeed averaged over that 5-day window. We have clarified this in the caption.

4. Fig.1 (Inset): *Is it the annual mean maximum temperature or the annual maximum value of the daily maximum temperature?*

%% This is showing the annual maximum value of daily maximum temperature - we have clarified this in the figure caption.

5. *How many ensemble members out of the grand ensemble are capable of reaching close to reality?*

%% 28 members across the 4 lead times exceed the maximum value. However, if we relax this constraint and count the number of member that are within 5C of the value (which would still place them above the previous record annual maximum temperature), we obtain 130 members.

6. *If the plotted surface temperature and geopotential anomalies indicate 26-30 Jun 2021 conditions as in the top panel of Fig.1, the lead times for the 1 May forecast should be less than 2 months.*

%% We fully agree that this caption was unclear. The maps show the temperature anomalies on the date that the temperatures reached that closest predicted temperature within each ensemble member - so these dates do not necessarily align with the 26-30th June. We have revised the caption to clarify this. As in the seasonal forecast we use the entire summer season,

then this lead time could be between 1-4 months. We have corrected this mistake, thank you for noting it.

7. P15: *remove space between numbers and the deg-C sign.*

%% We have done so.

References

Zarzycki, C. M., & Jablonowski, C. (2015). Experimental Tropical Cyclone Forecasts Using a Variable-Resolution Global Model. *Monthly Weather Review*, 143(10), 4012–4037.
<https://doi.org/10.1175/MWR-D-15-0159.1>

McKinnon, K. A., & Simpson, I. R. (2022). How Unexpected Was the 2021 Pacific Northwest Heatwave? *Geophysical Research Letters*, 49(18), e2022GL100380.
<https://doi.org/10.1029/2022GL100380>

Myhre, G., Alterskjær, K., Stjern, C. W., Hodnebrog, Ø., Marelle, L., Samset, B. H., Sillmann, J., Schaller, N., Fischer, E., Schulz, M., & Stohl, A. (2019). Frequency of extreme precipitation increases extensively with event rareness under global warming. *Scientific Reports*, 9(1), Article 1.
<https://doi.org/10.1038/s41598-019-52277-4>

Best regards,

Nicholas J. Leach and co-authors

REVIEWERS' COMMENTS

Reviewer #2 (Remarks to the Author):

This is my second review of this article. The authors have adequately addressed my previous comments and made significant revisions to the manuscript. I have no further remarks on this version, and as such, I recommend it be accepted for publication.